# Waning Humoral Response after COVID-19 mRNA Vaccination in Maintenance Dialysis Patients and Recovery after a Complementary Third Dose

**DOI:** 10.3390/vaccines10030433

**Published:** 2022-03-11

**Authors:** Bogdan Biedunkiewicz, Leszek Tylicki, Waldemar Ślizień, Monika Lichodziejewska-Niemierko, Małgorzata Dąbrowska, Alicja Kubanek, Sylwia Rodak, Karolina Polewska, Piotr Tylicki, Marcin Renke, Alicja Dębska-Ślizień

**Affiliations:** 1Department of Nephrology, Transplantology and Internal Medicine, Medical University of Gdańsk, 80-210 Gdańsk, Poland; bogdan.biedunkiewicz@gumed.edu.pl (B.B.); lichotek@gumed.edu.pl (M.L.-N.); karolina.polewska@gumed.edu.pl (K.P.); piotr.tylicki@gumed.edu.pl (P.T.); adeb@gumed.edu.pl (A.D.-Ś.); 2NZOZ Diaverum, 81-519 Gdynia, Poland; waldemar.slizien@diaverum.com (W.Ś.); sylwia.rodak@diaverum.com (S.R.); 3Central Clinical Laboratory, University Clinical Centre, 80-952 Gdańsk, Poland; m.dabrowska@uck.gda.pl; 4Department of Occupational, Metabolic and Internal Diseases, Medical University of Gdańsk, 81-519 Gdynia, Poland; alicja.kubanek@gumed.edu.pl (A.K.); mrenke@gumed.edu.pl (M.R.)

**Keywords:** COVID-19, SARS-CoV-2, maintenance dialysis patients, mRNA vaccines, humoral immunity, seroconversion

## Abstract

The aim of this study was to analyze the waning of anti-spike (S) antibodies after mRNA vaccination against COVID-19 in maintenance dialysis patients, and to assess the safety and effectiveness of the complementary third dose. This was a prospective, longitudinal study in which we analyzed the kinetics of antibodies up to six months after a two-dose vaccination (first protocol) in infection-naïve dialysis patients (IN-Ds), previously infected dialysis patients (PI-Ds) and subjects without chronic kidney disease (the controls), as well as their humoral response to the third dose of the same mRNA vaccine (second protocol). The respective reduction in antibody titer after 3 and 6 months by 82.9% and 93.03% in IN-Ds (*n* = 109), 73.4% and 93.36% in PI-Ds (*n* = 32) and 75.5% and 88.8% in the controls (*n* = 20) was demonstrated. Consequently, a protective antibody titer above 141 BAU/mL was found in only 47.7% and 23.8% of IN-Ds after 3 and 6 months, respectively. After the third vaccine dose, a significant increase in antibody titer was observed in all groups, with increases by a factor of ×51.6 in IN-Ds, ×30.1 in the controls and ×8.4 in PI-Ds. The median antibody titer after the third dose differed significantly between groups, and was the highest in PI-Ds: PI-Ds, 9090 (3300–15,000) BAU/mL; the controls, 6945 (2130–11,800); IN-Ds, 3715 (1470–7325) (*p* < 0.001). In conclusion, we observed similar degrees of antibody waning in all patients. After 3 months, over half of the infection-naïve dialysis patients had a very low antibody titer, and almost twenty percent of them had no antibodies at all. The humoral response to the third dose was very good, raising their titer of antibodies to a higher level than those in the general population who have received the primary two-dose scheme. The results support the administration of a complementary third dose of the mRNA vaccine for dialysis patients as soon as possible.

## 1. Introduction

Vaccinations significantly reduced the mortality associated with the COVID-19 pandemic in the general population [1]. The neutralizing antibodies produced as a result of immunization act as a shield that prevents or significantly reduces the spread of the virus in the body. However, it is also known that their neutralizing antibodies disappear over time, resulting in a gradual reduction in their serum antibody titer [2]. This, in turn, creates a risk of the virus breaking down the immune barrier and developing into a breakthrough infection [3]. Booster doses of vaccines restore vaccine effectiveness by increasing the neutralizing antibody titer and improving its efficacy against variants of the virus [4].

Patients with chronic kidney disease that are dependent on dialysis are among the populations with the highest risk of death from COVID-19 [5]. Their 28-day probability of death before the start of population vaccinations was 25% for all hemodialyzed patients and 33.5% for those who were admitted into hospitals, according to a European Renal Association COVID-19 Database (ERACODA) report [6]. In our previous studies, we showed the extremely high mortality of COVID-19 for hemodialyzed patients from North Poland, with a fatality rate of up to 43.81% in the oldest subjects, and found that the most important factor determining poor prognosis was their frailty [7,8]. Unfortunately, dialyzed patients are overlooked in large clinical trials; hence, data on preventing the effects of vaccination and prior SARS-CoV-2 infection in this population appear with some delay and are not so numerous. It is known that almost all dialyzed patients respond to vaccination with the production of neutralizing antibodies [9,10]. However, their humoral response is weaker than in the general population [11]. The exception, in this respect, is individuals with prior SARS-CoV-2 infections, who have titers after vaccination that are many times higher than infection-naive subjects [9]. The sparse data on the waning of antibodies in the dialysis patient population shows inconsistent results [12,13]. The response to the third complementary dose of the mRNA vaccine, which is recommended on a regular schedule for immunocompromised individuals in many countries, appears to be very good [14,15]. To shed more light on these issues, we examined these topics in a controlled study of dialyzed patients and subjects without chronic kidney disease.

## 2. Materials and Methods

### 2.1. Study Design

This is a prospective, longitudinal observational study conducted in two protocols with two different cohorts. The study was approved by the Ethics Committee of the Medical University of Gdańsk (Resolution NKBBN/167/2021) and conducted in accordance with the Declaration of Helsinki.

#### 2.1.1. First Protocol

In the first protocol, we analyzed the kinetics of SARS-CoV-2 anti-spike (S) IgG antibodies up to six months after a two-dose mRNA vaccination against COVID-19, and compared the durability of the humoral response between infection-naïve dialysis patients (IN-Ds), previously infected (with SARS-CoV-2) dialysis patients (PI-Ds) and subjects without chronic kidney disease (the control). Serum samples for anti-S antibodies were obtained 14–16 days after the second dose of the mRNA vaccine, and after 3 months and 6 months. We compared the proportion of patients who maintained an anti-S antibody titer above the cutoff point for anti-S seroconversion (>33.8 BAU/mL) and the proportion whose anti-S antibody titer was greater than 141 BAU/mL, a concentration that provides 89.3% protection against SARS-CoV-2 infection in immunocompetent patients [16]. Patients with breakthrough SARS-CoV-2 infections confirmed during this period were excluded from the study. Nucleocapsid (N)-specific IgG antibody serostatus was checked to exclude asymptomatic SARS-CoV-2 infections.

#### 2.1.2. Second Protocol

In the second protocol, we compared the titers of anti-S IgG antibodies for IN-Ds, PI-Ds and the controls after the third complementary dose of an mRNA vaccine. Serum samples for anti-S antibodies were obtained before and 14–16 days after the third dose of the vaccine. Solicited common and expected adverse reactions shortly (i.e., within 7 days) after the third dose of the vaccine (reactogenicity), and unsolicited and serious adverse events, i.e., those reported by the participants without prompts from the medical staff or those observed by their physicians 1 month after the third dose, were analyzed as well.

### 2.2. Study Population

The dialysis cohort in the first protocol consisted of patients chronically dialyzed at our institutions with hemodialysis or peritoneal dialysis and vaccinated against COVID-19 with two doses of an mRNA vaccine, BNT162b2 (Comirnaty, Pfizer/BionTech), given according to the manufacturer’s recommendations. The study included patients who agreed to participate and whose serum was collected after the second dose of vaccine and after 3 months and 6 months. Patients with a known prior SARS-CoV-2 infection were also vaccinated according to the same rules and were enrolled. Control patients were included if they had a confirmed estimated glomerular filtration rate (eGFR) >60 mL/min, had not been confirmed with a SARS-CoV-2 infection and were vaccinated against COVID-19 with the same vaccines and schedule as the dialyzed patients.

The dialysis cohort in the second protocol consisted of dialyzed patients vaccinated against COVID-19 with two doses of an mRNA vaccine, either BNT162b2 (Comirnaty, Pfizer/BionTech) or mRNA-1273 (Moderna), given according to the manufacturer’s recommendations, who also received the third complementary dose of the mRNA vaccine six months after the second dose. Subjects received a third dose that was the same vaccine type as in the primary vaccination. The cohort included some patients from the first cohort who consented to a third vaccination, and other patients in whom antibody kinetics had not been monitored in the first protocol but who were vaccinated with a third dose. The control group consisted of the same control patients as in the first protocol, who received a third dose according to the same rules as the dialyzed patients.

### 2.3. Procedures and Analytical Methods

Quantitative determination of specific IgG antibodies to trimeric S-proteins as an indicator of the humoral response to vaccination was performed with a commercial chemiluminescent immunoassay kit (The LIAISON^®^ SARS-CoV-2 TrimericS IgG test, DiaSorin, Italy). The assay presents a sensitivity of 98.7% and a specificity of 99.5%, and agreement with neutralization in microneutralization tests: sensitivity/positive percent agreement (PPA), 100%; specificity/negative percent agreement (NPA), 96.9% [17]. Samples were interpreted as positive (seroconversion) or negative (no seroconversion) with a cutoff index value of >33.8 BAU/mL, in accordance with the manufacturer. N-specific IgG antibodies were assessed with a commercial chemiluminescent immunoassay kit (SARS-CoV-2 IgG, Abbott Laboratories, Chicago, IL, USA) to exclude those with a prior SARS-CoV-2 infection and to confirm breakthrough infections after vaccination. The N protein is present in the viral core and plays a vital role in viral transcription. Natural exposure induces a dominant antibody response against the N protein, but since the N proteins is not in the vaccine, there is no vaccine-induced response against it. Therefore, it can be a specific indicator of SARS-CoV-2 infection [18].

Reactogenicity data was obtained through interviews performed by health staff according to a standardized questionnaire, as described previously [19]. The grading scales were derived from the FDA Center for Biologics Evaluation and Research (CBER) guidelines on toxicity grading scales for healthy adult volunteers enrolled in preventive vaccine clinical trials. The assessments included solicited local reactions (pain, redness, swelling) and systemic reactions (fever, fatigue, headache, chills, vomiting, diarrhea, new or worsened muscle pain and new or worsened joint pain). Serious adverse events were defined as any untoward medical occurrence that resulted in death, was life-threatening, required inpatient hospitalization or the prolongation of existing hospitalization or resulted in persistent disability/incapacity.

### 2.4. Statistical Analyses

Continuous data was expressed as medians (interquartile ranges, IQR). Categorical variables are presented as counts (percentages). Continuous variables were first tested for normal distribution using the Shapiro-Wilk test, and then compared by the t-test if normally distributed, or by the Mann-Whitney or Wilcoxon tests, where appropriate, if non-normally distributed. Differences in continuous independent variables measured more than twice were assessed using the Kruskal-Wallis H test, with a multiple range test for paired comparisons. Differences in continuous paired variables measured more than twice were assessed using the Friedman test. The chi-square test was used for categorical variables. The association between two variables and the direction of their relationship was assessed by the Spearman correlation coefficient. Multivariable analysis of ANCOVA was performed to further explore the potential impact of the patients’ age on the observed differences in crude comparisons. In order to meet the assumptions for ANCOVA modeling (with a post-hoc Tukey test), continuous variables were square-root transformed prior to the analysis. The final model included anti-S IgG titer, age, BMI, sex, CCI score (mild vs. moderate-to-severe) and study group (IN-D vs. PI-D). The data were analyzed with Statistica (version 12.0, Stat Soft, Inc., Dell Software, Tulsa, OK, USA). *p* Values of <0.05 were considered statistically significant.

## 3. Results

### 3.1. Patient Characteristics

A total of 141 dialysis patients (134 hemodialyzed and 7 treated with peritoneal dialysis) were enrolled in the first protocol. The cohort was stratified based on evidence of a previous SARS-CoV-2 infection into 109 IN-D patients and 32 PI-D patients. The control group included 20 infection-naïve individuals without chronic kidney disease. Their characteristics are provided in Table 1. The patients did not differ with respect to sex, BMI and dialysis vintage. The control subjects had a significantly lower CCI index and age. Patients in all groups were vaccinated with BNT162b2. Seroconversion in anti-S IgG antibodies after the primary two-dose vaccination was observed in 20/20 (100%) of the controls, 32/32 (100%) of PI-D patients and 105/109 (96.3%) of IN-D patients. Anti-S IgG antibody titer after the second dose of vaccination is presented in Table 1.

A total of 139 dialysis patients (129 hemodialyzed and 10 treated with peritoneal dialysis) were enrolled in the second protocol. The cohort was stratified based on evidence of a previous SARS-CoV-2 infection into 104 IN-D patients and 35 PI-D patients. The control group included 20 individuals without chronic kidney disease. Patients did not differ with respect to sex, BMI, dialysis vintage or the time between vaccination and blood sampling. The control subjects were significantly younger, and had a significantly lower CCI index. Their characteristics are provided in Table 2.

### 3.2. Waning Anti-S IgG Antibodies after 3 and 6 Months (First Protocol)

In IN-D patients, anti-S antibody titer decreased by 82.9% and 93.03% after 3 months and 6 months, respectively (Figure 1; Table 1. *p* < 0.001). Antibody titer remained above the cutoff point for seroconversion in 82.6% (100 of 109) at 3 months and 67.9% (74 of 109) at 6 months. Respectively, 47.7% (52 of 109) and 23.8% (26 of 109) maintained an antibody titer greater than 141 BAU/mL after 3 and 6 months, a concentration that provides 89.3% protection in immunocompetent patients (Table 3).

In the controls, anti-S antibody titer decreased by 75.5% and 88.8% after 3 months and 6 months, respectively (Figure 1; Table 1. *p* < 0.001). 100% (20 of 20) of patients remained seropositive and 95% (19 of 20) maintained an antibody titer greater than 141 BAU/mL after 3 months. 95% (19 of 20) of patients remained seropositive and 70% (14 of 20) maintained an antibody titer greater than 141 BAU/mL after 6 months (Table 3).

In PI-D patients, anti-S antibody titer decreased by 73.4% and 93.6% after 3 months and 6 months, respectively (Figure 1; Table 1. *p* < 0.001). 100% (32 of 32) of patients remained seropositive and 96.9% (31 of 32) maintained an antibody titer greater than 141 BAU/mL after 3 months. 96.9% (31 of 32) of patients remained seropositive and 93.7% (30 of 32) maintained an antibody titer greater than 141 BAU/mL after 6 months (Table 3).

Three months after the second dose, median anti-S antibody titer differed significantly between groups: PI-Ds, 2945 (1600–11,500) BAU/mL; the controls, 508 (422–1127); IN-Ds, 159 (42.3–357) (*p* < 0.001). Six months after the second dose, median anti-S antibody titer differed significantly between groups: PI-Ds, 723 (474–1100); the controls, 231 (103–336); IN-Ds, 66 (24–127) (*p* < 0.001). Detailed results are presented in Table 1.

### 3.3. Anti-S IgG Antibody Titer after the Third Vaccine Dose (Second Protocol)

After the third dose, a significant increase in anti-S antibody titer was observed in all groups by a factor of ×51.6 (IN-Ds), ×30.1 (the controls) and ×8.4 (PI-Ds) (*p* < 0.001 for each group) (Table 2). Individually, 100% (35 of 35) of PI-Ds, 100% (20 of 20) of the controls and 95.2% (99 of 104) of IN-Ds developed antibody titers greater than 141 BAU/mL. There was no anti-S seroconversion in any of the four IN-D patients who did not respond to the prime two-dose vaccination.

Median anti-S antibody titer after the third dose differed significantly between groups: PI-Ds, 9090 (3300–15,000) BAU/mL; the controls, 6945 (2130–11,800); IN-Ds, 3715 (1470–7325) (*p* < 0.001) (Figure 2). The pairwise comparisons between groups with the multiple range test showed the following significance: (IN-D vs. PI-D: *p* = 0.001). It was also confirmed in a multivariable approach of ANCOVA where the difference in anti-S antibodies titers between PI-Ds and IN-Ds were significant after adjusting for confounders (*p* < 0.01 for the model; *p* < 0.001 post-hoc Tukey test) (Appendix A in Appendix A). In strata analyses performed on the IN-D group, there were no differences in anti-S antibody titer after the third dose by age, BMI, gender, comorbidity index and vaccine type, or between peritoneal dialysis and hemodialysis patients (Table 4). There was a positive correlation between antibody titer after the second dose and after the third dose (r = 0.403, *p* < 0.05) (Figure 3). Comparing the anti-S IgG titer after the second and third doses within each subgroup, the titer after the third dose was significantly higher compared to the titer after the second dose in IN-Ds (*p* < 0.001) and the controls (*p* < 0.001), and was lower in PI-Ds (*p* = 0.47). Antibody titer in IN-Ds after the third dose was significantly higher compared to the titer after the second dose in the controls (*p* = 0.026) (Figure 2).

### 3.4. Reactogenicity to the Third Vaccine Dose

Of 88 IN-D patients (16 patients did not respond), 57.9% reported at least one local site reaction within 7 days after the third dose of the mRNA vaccine. They reported only mild-to-moderate injection site reactions. No grade 3 or 4 local reactions were reported. Pain at the injection site was the most frequent local reaction among the vaccines. The median duration of local reactions was 2.25 days. At least one solicited systemic reaction occurred in 21.6% of IN-Ds. The most frequent solicited systemic reactions were fatigue (17.0%), followed by muscle pains (12.5%), chills (11.4%) and fever (7.9%). The majority of patients reported only mild-to-moderate systemic reactions. One patient (1.1%) had severe systemic symptoms in the form of a high fever. No grade 4 systemic reactions were reported. The median duration of systemic symptoms was 1 day. No serious adverse events following the vaccination were reported. Of 17 control patients (3 patients did not respond), 64.7% reported at least one local site reaction within 7 days after the third dose of the mRNA vaccine. They reported only mild-to-moderate injection site reactions. Pain at the injection site was the most frequent local reaction to the vaccine At least one solicited systemic reaction occurred in 35.3% of subjects. The most frequent solicited systemic reactions were fatigue (23.5%), followed by muscle pains (11.8%), headache (11.8%), chills (5.9%) and fever (5.9%). The majority of patients reported only mild-to-moderate systemic reactions. One patient (6.2%) had severe systemic symptoms in the form of high fever. No grade 4 systemic reactions were reported. No serious adverse events following the vaccination were reported. The incidence of local site and systemic side effects did not differ between the studied groups.

## 4. Discussion

Previous studies have shown that dialysis patients respond to vaccination against COVID-19, although the immune response is significantly weaker than in patients from the general population [11]. However, the kinetics of antibody waning after vaccination is poorly understood in this group of patients. A few studies indicate a faster decline in the neutralizing antibody titer in dialysis patients than in the general population [20]. It is not fully known what vaccination schedule against COVID-19 is optimal for dialysis patients, and at what number of doses and in what intervals it should be administered. The U.S. Food and Drug Administration, the Centers for Disease Control and Prevention and many national health system organizations recommend the third complementary dose as a regular course of vaccination with mRNA vaccines for immunocompromised individuals.

Our study showed that the rate of antibody waning is similar in the general population and in both groups of dialysis patients. After 6 months, the decline in antibody titer was about 90% in all groups. Similarly, a decrease in IgG anti-S titer by 89.6% at 6 months after vaccination with BNT162b2 was reported by Bayart et al. in healthcare workers without chronic kidney disease [21], while a 92.3% reduction was observed in a small population of hemodialysis patients in a study by Davidovic et al. [12]. The differences in antibody titer between groups at 3 months and 6 months resulted from the titer threshold being reached by the patients after the second dose of their primary vaccine. In the group with the weakest response, that is, dialyzed patients without prior infection, the percentage of seropositive subjects decreased to almost 68% after 6 months. In a study with a shorter follow-up, Speer et al. reported that seropositivity for anti-S1 IgG antibodies decreased after 3 months from 95% to 88% in peritoneal dialysis patients, and from 88% to 77% in hemodialysis patients [22]. Others indicated that 6 months after vaccination, the seroconversion rate, similar to our study, was only 65.8% [12]. In a quite recent study of a U.S. national cohort of patients receiving dialysis, Anand et al. reported that 20% of subjects had lost a detectable antibody response within 6 months after vaccination. In addition, low levels of circulating receptor-binding domain antibodies were associated with a risk for breakthrough infection [13]. In our study, more than half of infection-naïve dialysis patients have a low antibody titer below 141 BAU/mL, and as a consequence, they are probably not protected from infection only 3 months after the prime vaccination [16]. This confirms the need to give the third complementary dose of the vaccine to dialyzed patients as soon as possible after the standard two-dose vaccination.

This study is one of the first on dialysis patients to show a significant increase in antibody titer after the third dose of the mRNA vaccine [14,15,23]. Although the antibody titer in infection-naïve dialysis subjects turned out to be lower than in the healthy and dialyzed convalescents, similar to what was observed earlier in the case of the primary cycle of vaccination [9], it was three times higher than after the second dose in the same group, and almost twice as high as in the control subjects after the second dose. The results are all the more promising as the dialyzed patients were significantly older than the control subjects in our study. The good humoral response of the dialyzed patients to the third dose is also confirmed by other studies with a smaller sample size. For example, Dekervel et al. showed that the strength of the immune response to the third dose of an mRNA vaccine in hemodialyzed patients is similar to the response of healthy individuals to the two-dose primary vaccination [23]. Several studies have shown that the third dose of the vaccine produces a humoral response in half or most of the dialyzed patients who were not seroconverted after the two-dose primary vaccination [15,24]. Others demonstrated that patients with a poorer response to the primary vaccination had higher antibody production after the third dose than those with a higher antibody titer [14,25]. Our study failed to confirm such a relationship. On the contrary, we observed a positive correlation between the antibody titer after the second and third dose of the vaccine. Similar to the study of Dekervel et al. [23], and in contrast to Tillmann’s findings [15], we did not show an age-response relationship to the third dose of the vaccine, which is commonly seen in the primary two-dose vaccination [9,26,27]. Primary vaccination non-responders did not react to a third vaccination in our study. However, it was a very small group of patients, consisting of only four individuals; hence, the conclusions are of limited value in this respect.

Once again, it has been shown that earlier SARS-CoV-2 infection (additional natural immunization) has a positive effect on humoral response after vaccination [9,28,29,30]. The antibody titer after the third dose in PI-D patients was three times higher than that of IN-D subjects. However, unlike IN-Ds, the antibody titer after the third dose in PI-D subjects is lower than after the second dose of the vaccine in the same patients. Perhaps too-frequent immunization (natural plus triple vaccine immunization) impairs the strength of the humoral response to an antigenic stimulus to some extent, as a result of the phenomenon known as T-cell anergy or exhaustion, which is observed in cancer or chronic viral infections where antigenic stimulation occurs repeatedly [31,32]. This may lead to further studies on the optimal regimen (mRNA dose and vaccination intervals) for this group of patients.

## 5. Limitations

Differences in age between the study and control patients could represent a major confounding factor, since it is well documented that, with increasing age, there is a reduction in antibody responses after vaccinations [33]. At the time of our analysis, the booster doses were given only to healthcare workers and immunocompromised persons. Taking this and the observational nature of our work into account, it was impossible to avoid this limitation. However, it should be emphasized that our analyzes did not show any significant influence of age on the humoral response after the third dose in our population. Moreover, multivariable analysis that included age as a variable confirmed independent differences in the strength of the humoral response to the third vaccination between the study groups.

## 6. Conclusions

We observed similar degrees of anti-S antibody waning after vaccination in dialyzed patients as in the general population. Low antibody titer after a prime vaccination in infection-naïve dialysis patients caused over half of them to have a very low antibody titer after only 3 months, which probably does not protect them from infection, with only one-fourth of them having antibodies at all. Vaccination with the third dose, considered complementary in this population, was well tolerated. The humoral response was very good, raising the level of antibodies to a higher level than in subjects from the general population that received the primary two-dose scheme. The results confirm the validity of administering complementary vaccinations to immunocompromised individuals as early as possible.

## Figures and Tables

**Figure 1 vaccines-10-00433-f001:**
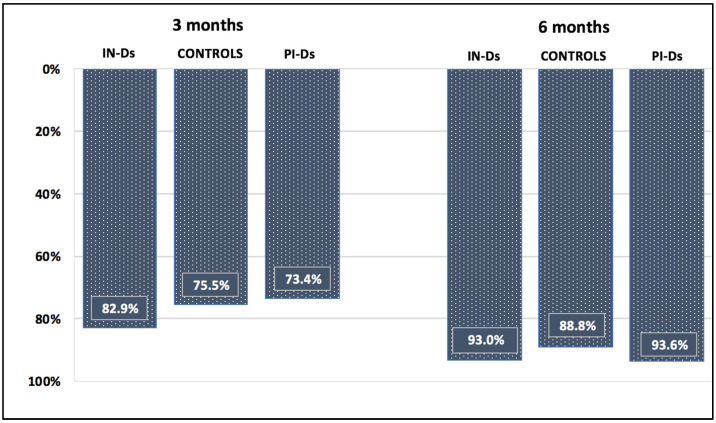
Anti-S IgG titer decline over time after second dose of vaccination.

**Figure 2 vaccines-10-00433-f002:**
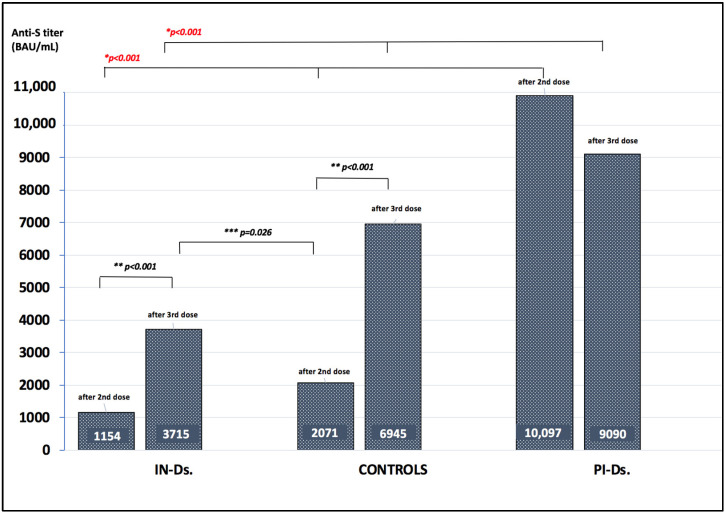
Anti-S antibody titer after the second and third dose of mRNA vaccine. * Kruskal-Wallis test: (IN-Ds vs. the controls vs. PI-Ds). Multiple range test for paired comparisons: after second dose, (IN-Ds vs. the controls: *p* = 0.002) and (IN-Ds vs. PI-Ds: *p* < 0.001); after third dose, (IN-Ds vs. PI-Ds: *p* = 0.001). Additional secondary analyses: ** *p* < 0.001 (IN-Ds’ second vs. IN-Ds’ third dose; controls’ second vs. controls’ third dose). *** *p* = 0.026 (IN-Ds’ third dose vs. controls’ second dose).

**Figure 3 vaccines-10-00433-f003:**
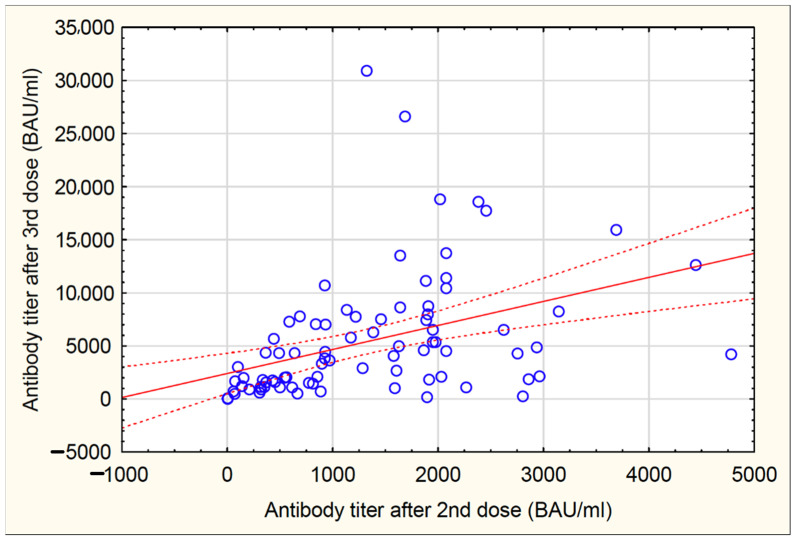
Association between anti-S antibody titer after second and third dose in IN-D patients.

**Table 1 vaccines-10-00433-t001:** Patient characteristics (protocol 1).

	IN-Ds N = 109	PI-Ds N = 32	Controls N = 20	*p*-Value *
Age years	69 (57–75)	65 (58–74)	53 (47–69.5)	0.09
Male sex Female sex	69 (63.30) 40 (36.70)	22 (68.75) 10 (31.25)	13 (65) 7 (35)	0.57 0.57
CCI	6 (4–8)	6.5 (4–8)	1 (1–3)	<0.001
Diabetes mellitus	40 (36.69)	6 (18.75)	0 (0)	0.059
BMI kg/m^2^	24.9 (22.4–29.3)	26.1 (23.2–29.1)	27 (24.5–30.5)	0.5
Dialysis vintage months	36 (15–74)	39 (19.5–84)	na	0.71
Anti-S IgG after 2nd dose (BAU/mL)	933 (528–1906)	10,907 (2502–18,031)	2070 (1703–3068)	<0.001
Anti-S IgG 3 months after 2nd dose (BAU/mL)	159 (42.3–357)	2945 (1600–11,500)	508 (422–1127)	<0.001
Anti-S IgG 6 months after 2nd dose (BAU/mL)	66 (24–127)	723 (474–1110)	231 (103–336)	<0.001

Abbreviations: CCI, Charlson comorbidity index; BMI, body mass index; na, not applicable. Data are expressed as medians (interquartile ranges (IQR)) for continuous variables and counts (percentages) for categorical variables. * Difference between IN-Ds vs. PI-Ds vs. controls (Kruskal-Wallis H test or chi^2^ test); between IN-Ds vs. PI-Ds (Mann-Whitney test).

**Table 2 vaccines-10-00433-t002:** Patient characteristics (protocol 2).

	IN-Ds N = 104	PI-Ds N = 35	Controls N = 20	*p*-Value *
Age years	70 (58.5–76)	62 (46–70)	53 (47–69.5)	0.002
Male sex Female sex	65 (62.5) 39 (37.5)	24 (68.57) 11 (31.43)	13 (65) 7 (35)	0.52 0.52
CCI	6.5 (4–8)	6 (3–7)	1 (1–3)	<0.001
Diabetes mellitus	36 (34.61)	8 (22.86)	0	0.019
BMI kg/m^2^	25.6 (22.7–29.2)	26.3 (22.86–28.1)	27 (24.5–30.5)	0.40
Dialysis vintage months	33.5 (11–68.5)	36 (10–60)		0.77
BNT162b2 vaccination mRNA-1273 vaccination Anti-S IgG after 2nd dose (BAU/mL)	96 (92.3) 8 (7.7) 1154 (474–1952)	30 (85.7) 5 (14.3) 10,907 (1342–13,754)	20 (100) 0 (0) 2070 (1703–3068)	0.40 0.40 <0.001
Anti-S IgG before 3rd dose (BAU/mL)	72 (25–160)	1080 (474–1660)	231 (102–336)	<0.001
Anti-S IgG after 3rd dose (BAU/mL)	3715 (1470–7325)	9090 (3300–15,000)	6945 (2130–11,800)	<0.001

Abbreviations: CCI, Charlson comorbidity index; BMI, body mass index. Data are expressed as medians (interquartile ranges (IQR)) for continuous variables and counts (percentages) for categorical variables. * Difference between IN-Ds vs. PI-Ds vs. controls (Kruskal-Wallis H test or chi^2^ test).

**Table 3 vaccines-10-00433-t003:** Proportion of patients with anti-S antibody titer above the seroconversion cutoff point and the 141 BAU/mL threshold.

	Follow-Up	IN-Ds	CONTROLS	PI-Ds	*p*-Value
Seroconversion titer > 33.8 BAU/mL	3 months	82.6%	100%	100%	0.05
	6 months	67.9%	95%	96.9%	<0.001
Protective titer > 141 BAU/mL	3 months	47.7%	95%	96.9%	<0.001
	6 months	23.8%	70%	93.7%	<0.001

**Table 4 vaccines-10-00433-t004:** Strata analyses of anti-S antibody (BAU/mL) titer after the second and third dose in IN-D patients.

	N (%)	Anti-S IgG 2nd Dose	*p*-Value	Anti-S IgG 3rd Dose	*p*-Value
Age < 70 years Age ≥ 70	56 (53.85) 48 (46.15)	1578 (637–1919) 926 (320–1919)	0.06	4005 (1640–6865) 3460 (1160–7565)	0.82
Male sex Female sex	65 (62.5) 39 (37.5)	1134 (455–1905) 1219 (494–2080)	0.56	3810 (1130–7250) 3490 (1640–7510)	0.92
CCI ≥ 6.5 CCI < 6.5	53 (50.96) 51 (49.04)	926 (330–1936) 1423 (629–2000)	0.19	3810 (1190–7400) 3490 (1520–7250)	0.94
BMI ≥ 25.6 kg/m^2^ BMI < 25.6	52 (50) 52 (50)	1175 (494–1908) 1133 (445–1953)	0.95	3650 (1600–7250) 3795 (1410–7325)	0.95
Diabetes No diabetes	36 (34.6) 68 (65.4)	932 (445–1498) 1336 (494–2033)	0.11	4630 (1160–7990) 3140 (1530–6755)	0.46
HD patients PD patients	94 (90.4) 10 (9.6)	933 (445–1918) 1641 (1578–2080)	0.08	3890 (1400–7250) 2975 (2080–8600)	0.65
BNT162b2 mRNA-1273	96 (92.3) 8 (7.7)	1154 (475–1952) no data	na	3890 (1410–7140) 2865 (1680–9595)	0.68

Abbreviations: CCI, Charlson comorbidity index; BMI, body mass index; HD, hemodialysis; PD, peritoneal dialysis. Data are expressed as medians (interquartile ranges (IQR)) for continuous variables and counts (percentages) for categorical variables. Stratification was performed against the median.

## Data Availability

Detailed data are available on request from the corresponding author.

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
