# Peer review of "Waning Humoral Response after COVID-19 mRNA Vaccination in Maintenance Dialysis Patients and Recovery after a Complementary Third Dose"

_vaccines, 2022, doi:10.3390/vaccines10030433_

Round 1

Reviewer 1 Report

Dear Authors, 
I have read with interest the article entitled "Waning humoral response after COVID-19 mRNA vaccination 2 in maintenance dialysis patients and recovery after a complementary third dose" and I think it could be of interest for Vaccines.
However there are some major revisions that you should consider as reported below:

1) In the abstract the size of each group should be reported;

2) The results in the abstract are confusing and they should be summarized in a more suitable way.

3) The last sentence in the abstract seems to be not supported by the results of the study;

4) In the Methods please add references with regard to the data reported about the commercial chemiluminescent immunoassay;

5) Sample size: how did you evalute the sample size of each group? There has been a matching between cases and controls? The difference in age could represent a major confounding factor since it is well documented that increasing the age there is a reduction in antibody response (Amodio et al Vaccines (Basel). 2021 Jul 1;9(7):714. doi: 10.3390/vaccines9070714; Bonura et al. J Appl Microbiol. 2022 Jan 26. doi: 10.1111/jam.15463). Are you sure that there is not a statistical difference in age between the three groups? Could please report, only in the rebuttal letter, the age of all patients?

Add your considerations to the Discussion section.

6) Statistical analyses: there a large number of analytical mistakes in p-value calculation and methodology. Please refer to a experienced statistician or the article will be not suitable for publication.

7) Statistical tests should be checked since they are erroneously reported (Firedman? - ANOVA Kruscal-Wallis?) 

8) A correlation analysis is reported in figure 2 without any mention in the statistical analyses.

9) Statistical analyses are very confused. Please: a) Comparison within groups and between groups should be reported in a matrix or in a other more clear representation; b) Given the characteristics of data, an adjustment for other variables (age, comorbidities) could be required by multivariable approach or otherwise consider to obtain three more comparable groups by excluding subjects with different profiles (very old; with several comorbidities etc);

10) In the discussion you report considerations about frequent immunization. However it is opinion of this Reviewer that your data do not support such considerations and the references could be not appropriate and sufficient for assuring the reported suggestions.

11) Please explain why you exclude that the lower titers in the IN-D groups can be related with the higher prevalence of diabetes or the older age of this group. This is a major limitation of the study that should be carefully evaluated (also through multivariable statistical analyses) in order to support (or rebut) the conclusion of your study.

Given the importance of all of the limitations, only if all the required points will be satisfied the paper could be considered suitable for publication.
Sincerely.

Author Response

Due to the numerous comments of both Reviewers (some identical) and the significant changes we have made to the manuscript - we present responses and comments to the review in one answer available to both Reviewers.

Reviewer 2 Report

Biedunkiewicz et al. conducted a clinical study to evaluate Covid mRNA vaccine-induced anti-S antibody responses in dialyzed patients with (PI-D) or without prior infection (IN-D) and non-dialyzed healthy patients (Control). The major findings include quick anti-S antibody declining in all groups of patients at a similar rate and the third dose could significantly enhance anti-S antibody levels. These findings are mostly in line with the literature (ref. 19-21) and the novelty of the study was not very clear. Since references (19-21) are highly related to the current study, the major findings need to be introduced in the discussion. I also have below comments to potentially help to improve the manuscript.

Major comments:

  1. The method mentioned two mRNA vaccines (Pfizer, Moderna) were used in the study. It’s important to analyze the results based on the vaccine type.
  2. Seroconversion was not defined (how much antibody increase).
  3. Figure 2 data were introduced in Table 2. If the purpose was to show more group differences, comparisons between groups after 3rd dose were also needed.
  4. Table 3 only included data after 3rd dose. It’s also important to include data after 2nd dose. Also, hemodialysis and peritoneal dialysis patients were not listed in Table 3.
  5. Reactogenicity in other groups than IN-D was also needed for a comparison in 3.3.
  6. Anti-S antibody titer was compared between 2nd and 3rd dose and found anti-S antibody titer was lower after 3rd dose in PI-D group than 2nd dose. This led authors to question frequent immunization may not be good for PI-D patients. Such an illustration may need to be revised considering 3rd dose still significantly increased serum anti-S antibody titer (from 1080 before 3rd dose) to a level (9090), which was slightly lower than after 2nd dose (10907) and still considered protective.

Minor comments:

  1. Anti-s and Anti-S were both used and need to be consistent.
  2. The secondary endpoints were introduced. Are there primary endpoints for this study?
  3. In 2.3, PPA and NPA were not defined. Other abbreviations also need to be checked to see whether they were defined at first appearance.
  4. Several P values were missing in Table 2.
  5. Grammatical and typo errors need to be checked and corrected.

Author Response

(The authors gave the same response as above.)

Round 2

Reviewer 1 Report

Dear Authors, 
I have read with interest the revised version of the manuscript and I think it has been significantly improved.
However, before publication some minor revisions will need as reported below:

1) In the abstract, the square bractets should be erased;

2) The abstract is still very confused in the result part. Please improve it since it is one of more important part of the manuscript;

2) Please clarify what kind of multivariable analysis have you performed;

3) Please, report the mean difference (in days) between vaccination and blood sampling for each group and evaluate if this value has significant diffence between groups;

4) in Results you state that "Patients in all 174 groups were vaccinated with BNT162b2 but this seems to be in contrast with that stated in material and methods. Please clarify why and check;

5) In table 4 percentages should be added also for Vaccine type;

6) In Results you report that " It was also confirmed in multivariable approach where difference in anti-S antibodies titers between PI-D and IN-D were significant after adjusting for confounders (P<0.01 for the model; P<0.001 post-hoc Tukey test). The Results shoud be reported in depth by describing the model (in Material and Methods) and the confounders. It could be better to add a table for summarize the results.

Author Response

Answer for reviewers in the attached file.

Reviewer 2 Report

Authors addressed most of the comments. A few minor format changes can be made to improve the reading. For example, table titles are usually put on top of the table and figure titles are usually put under the figure. The two (p<0.001) labels in figure 2 are confusing about the comparison groups.  

Author Response

(The authors gave the same response as above.)
